# Incidence, Mortality, and Trends of Prostate Cancer in Mexico from 2000 to 2019: Results from the Global Burden of Disease Study 2019

**DOI:** 10.3390/cancers14133184

**Published:** 2022-06-29

**Authors:** Saul A. Beltran-Ontiveros, Martha A. Fernandez-Galindo, Jose M. Moreno-Ortiz, Jose A. Contreras-Gutierrez, Jesus Madueña-Molina, Eliakym Arambula-Meraz, Emir Leal-Leon, Delia M. Becerril-Camacho, Veronica J. Picos-Cardenas, Carla Angulo-Rojo, Diana Z. Velazquez, Francisco Jimenez-Trejo, Francisco Gallardo-Vera, Daniel Diaz

**Affiliations:** 1Posgrado en Ciencias Biomédicas, Facultad de Ciencias Químico Biológicas, Universidad Autónoma de Sinaloa, Culiacán Rosales 80010, Sinaloa, Mexico; saul.beltran@uas.edu.mx; 2Doctorado en Genética Humana, Departamento de Biología Molecular y Genómica, Universidad de Guadalajara, Guadalajara 44340, Jalisco, Mexico; 3Instituto de Genética Humana “Dr. Enrique Corona Rivera”, Centro Universitario de Ciencias de la Salud, Universidad de Guadalajara, Guadalajara 44340, Jalisco, Mexico; jmiguel.moreno@cucs.udg.mx; 4Centro de Investigación y Docencia en Ciencias de la Salud, Universidad Autónoma de Sinaloa, Culiacán Rosales 80030, Sinaloa, Mexico; drcontreras9@hotmail.com; 5Facultad de Medicina, Universidad Autónoma de Sinaloa, Culiacán Rosales 80246, Sinaloa, Mexico; investigacion.cidocs@uas.edu.mx (J.M.-M.); veronicapicos@uas.edu.mx (V.J.P.-C.); carla.angulo@uas.edu.mx (C.A.-R.); 6Laboratorio de Genética y Biología Molecular, Facultad de Ciencias Químico Biológicas, Universidad Autónoma de Sinaloa, Culiacán Rosales 80010, Sinaloa, Mexico; eliakymarambula@hotmail.com (E.A.-M.); emir.leal@uas.edu.mx (E.L.-L.); 7Laboratorio de Biomedicina, Universidad Autónoma de Occidente, Unidad Regional Culiacán, Culiacán Rosales 80020, Sinaloa, Mexico; deliabecerril76@gmail.com; 8Departamento de Biomedicina Molecular, Centro de Investigación y de Estudios Avanzados, Gustavo A. Madero 07360, Ciudad de México, Mexico; velazquez.mvz@gmail.com; 9Laboratorio de Morfología Celular y Tisular, Instituto Nacional de Pediatría, Coyoacán 04530, Ciudad de México, Mexico; trejofj@hotmail.com; 10Laboratorio de Biología Molecular y Bioseguridad Nivel III, Centro Médico Naval, Coyoacán 04470, Ciudad de México, Mexico; jfgallardo81@gmail.com; 11Centro de Ciencias de la Complejidad (C3), Universidad Nacional Autónoma de México, Coyoacán 04510, Ciudad de México, Mexico

**Keywords:** age-standardized rate, burden of disease, cancer epidemiology, malign neoplasm, sociodemographic index, subnational heterogeneity

## Abstract

**Simple Summary:**

Worldwide, prostate cancer (PC) causes high morbidity and mortality. Thus, developing effective strategies for the prevention, diagnosis, and control of this disease is fundamental to providing updated and reliable estimations of the PC burden both nationally and subnationally. Herein, we analyzed data from the Global Burden of Disease study to estimate PC incidence and mortality, and their trends in Mexico at the national and subnational levels from 2000 to 2019. Our results show that PC was the top ranked cause of death among malign neoplasms in males from Mexico during 2019. Males from 70 to 79 years of age were the most affected by PC, and there was an increasing trend in the burden of this cancer. There was substantial subnational heterogeneity that suggested differential geographical patterns of change. These results provide both comprehensive and comparable estimates to assist the effort to reduce health loss due to PC.

**Abstract:**

In 2019, the Global Burden of Disease (GBD) estimated that prostate cancer (PC) was the 16th most common cause of death globally in males. In Mexico, PC epidemiology has been studied by a number of metrics and over various periods, although without including the most up-to-date estimates. Herein, we describe and compare the burdens and trends of PC in Mexico and its 32 states from 2000 to 2019. For this study, we extracted online available data from the GBD 2019 to estimate the crude and age-standardized rates (ASR per 100,000 people) of the incidence and mortality of PC. In Mexico, PC caused 27.1 thousand (95% uncertainty intervals, 20.6–36.0 thousand) incident cases and 9.2 thousand (7.7–12.7 thousand) deaths in males of all ages in 2019. Among the states, Sinaloa had the greatest ASR of incidence, and Guerrero had the highest mortality. The burden of PC showed an increasing trend, although the magnitude of change differed between metrics and locations. We found both an increasing national trend and subnational variation in the burden of PC. Our results confirm the need for updated and timely estimates to design effective diagnostic and treatment campaigns in locations where the burden of PC is the highest.

## 1. Introduction

Prostate cancer (PC) is a major cause of morbidity and mortality in men because it is the second most frequently diagnosed cancer worldwide [1]. According to the most recent iteration of the Global Burden of Disease study (GBD), in 2019, the age-standardized rate of PC was globally ranked as the 16th most common cause of death for males [2]. During this same year, there were an estimated 1.41 million cases and 486,836 deaths due to PC [3]. Increased public awareness and the ease of prostate-specific antigen (PSA) testing have led to PC being considered a public health priority in low- and middle-income settings [4]. In Latin America, the burden of PC follows the global trend [5], although with the existence of heterogeneity among the countries from this region [6]. This heterogeneity exposes the necessity for a further detailed characterization of the PC burden by country, as this information will provide a better overview of the patterns and trends of this disease.

More specifically, PC has become a public health problem in Mexico [7], where in 2019, this malign neoplasm was ranked as the 4th most common cause of death by cancer in males of all ages [3]. Even though previous studies have described the burden of PC in Mexico, these have had a narrow scope that included only a limited number of years, population coverage, localities, and epidemiological metrics, thereby providing geographical coverage that is either partial or too broad regarding the burden and trends of this disease [5,8,9]. In addition, a malignant tumor follow-up registry is absent in Mexico [10]. Consequently, timely and accurate epidemiological estimates represent an opportunity to provide updated and reliable information for effective health system planning and appropriate resource allocation. Herein, we use estimates from the GBD 2019 to present a detailed and comparative epidemiological description of the burden and trends of PC in Mexico at the national and subnational levels by age group from 2000 to 2019.

## 2. Methods

### 2.1. Data Input Sources and Case Definition

To estimate the burden of PC in Mexico, the GBD used 166 data input sources that corresponded to Vital Registration Death Data, household surveys, censuses, and multiple indicator cluster surveys from the National Institute of Statistics and Geography (INEGI) available from 1980 to 2017. A detailed list and information regarding the data input sources used are available on the Global Health Data Exchange (GHDx) Data Input Sources Tool (http://ghdx.healthdata.org/gbd-2019/data-input-sources accessed on 10 January 2022). For this study, PC cases for incident and mortality data were identified as codes C61-C61.9, D07.5, D29.1, D40.0, Z12.5, Z80,42, and Z85.46 according to the International Code of Diseases 10th (ICD10); and codes 185–185.9, 222.2, 236.5, V10.46, and V76.44 for ICD9 [11].

### 2.2. Estimation of Prostate Cancer Incidence and Mortality

A detailed description of the PC incidence and mortality estimates for the GBD are already published [11]. The modeling framework and a detailed flowchart including specific codes for the cancer burden estimation in the GBD are available at http://ghdx.healthdata.org/gbd-2019/code/cod-2 (accessed on 12 January 2022). The modeling steps included (1) calculating the mortality-to-incidence ratio (MIR) using data sources that included incidence and mortality for PC, (2) collecting the incidence of PC for each cancer registry, (3) estimating PC mortality by multiplying the incidence data by the corresponding MIR, (4) incorporating PC mortality sets into the PC cause of death database using the Cause of Death Ensemble model (CODEm) and processing the data to estimate the cancer-specific mortality of PC, and (5) calculating the incidence of PC using the estimated cancer-specific mortality of PC and MIR [12]. Additionally, we calculated the national and subnational case fatality rates of PC (CFR = mortality/morbidity × 100) for 2019.

### 2.3. Reporting Standards

To describe the PC burden in Mexico, we used data publicly available online at https://vizhub.healthdata.org/gbd-results/ (accessed on 15 January 2022). We collected data at the national level and for 32 states, including Mexico City, from 2000 to 2019. For all estimates, uncertainty was propagated through each modeling step; 95% uncertainty intervals (UIs) represent the 2.5th and 97.5th percentiles of the distribution of 1000 random draws performed at each step [3]. All estimations were performed by location, year, and age group; 95% UIs are reported. To summarize the PC burden, we used the crude and age-standardized rate (ASR) per 100,000 people for the incidence and mortality from 2000 to 2019 in males. Data processing and analyses were conducted by the GBD using Python version 3.7.0 (Python Software Foundation); Stata version 15.1 (StataCorp); and R version 3.4.1 (R Foundation). The code is available at https://ghdx.healthdata.org/gbd-2019/code (accessed on 10 January 2022) [3].

### 2.4. Analysis of Prostate Cancer Trends

We assessed the trends of the PC burden by estimating the percentage change in crude and ASR of the incidence and mortality during the last decade (2010 to 2019). Additionally, we performed a jointpoint regression analysis to evaluate the temporal trend in the national ASR of incidence and mortality using Jointpoint regression software 4.9 (https://surveillance.cancer.gov/joinpoint/ accessed on 3 June 2022) from the National Cancer Institute. Based on ASR and their standard errors, the model calculated the annual percent change (APC) for any given segment and expressed the estimate assuming a constant rate during the time interval. Additionally, the analysis calculated the average annual percent change (AAPC) for the entire time interval. The APC and AAPC were estimated with 95% confidence intervals (95% CI) to assess both the direction and magnitude of change considering the value and sign of the slope for a given segment or the full period [13]. Finally, we assessed whether APC and AAPC varied significantly from zero by using a two-sided t-distribution test at the alpha level = 0.05.

### 2.5. Analysis of Prostate Cancer Burden by Age Group

To assess the specific pattern of the PC burden according to age group, only cases from 40 to >95 years were included. We included crude and ASR (per 100,000 people) estimates for the incidence and mortality due to PC per five-year age group. Using these groups, we plotted the death counts by year from 2000 to 2019 to assess the specific contribution of each group to mortality due to PC in Mexico. Finally, we compacted age groups into ten-year categories to assess their relative contributions to the incidence and mortality counts in 2019.

### 2.6. Association of Prostate Cancer Burden with Sociodemographic Index

We used the sociodemographic index (SDI), which represents a composite indicator that includes fertility, education, and income [14], to assess its association with the burden of PC. The SDI is stratified into quintiles and ranges from 0 to 1, indicating the theoretical minimum and maximum level of development relative to these relevant health outcomes. Based on the results from the Shapiro–Wilk test for the normal distribution (incidence, W = 0.96, *p* = 0.393; deaths, W = 0.954, *p* = 0.189), we performed a Pearson correlation analysis to assess the linear association between the subnational SDI and each metric. We considered *p* < 0.05 to be significant. All figures were constructed with Prism 9 (GraphPad).

## 3. Results

### 3.1. National Incidence and Mortality of Prostate Cancer in 2019

In Mexico, there were 27,096 (95% UI, 20.6 to 36.0 thousand) incident cases of PC and 9256 (95% UI, 7.7 to 12.7 thousand) deaths in 2019 for men of all ages, which resulted in a CFR of 34.2%. In the same year, the age-standardized incidence and mortality rates were estimated at 52.3 (40.0 to 70.1) cases and 19.4 (14.7 to 26.7) deaths per 100,000 people, respectively (Table 1). According to the age-standardized global ranking of the 45 groups of diseases and injuries included in level 3 of the GBD, with 15.3 (13.0 to 18.6) deaths per 100,000 people, PC was ranked as the 16th most common cause of death for males worldwide in 2019. In contrast, PC caused the 11th highest mortality rate in males from Mexico (Appendix A). Additionally, among the 27 cancer groups included in the GBD, PC was the top-ranked death rate per 100,000 people due to neoplasms in Mexico during 2019, whereas at the global level, PC was ranked 4th (Appendix A).

### 3.2. Subnational Incidence and Mortality of Prostate Cancer in 2019

At the subnational level, the highest number of incident cases in males of all ages during 2019 occurred in the state of Mexico (2.8 thousand, 2.1 to 4.2) and Mexico City (2.5 thousand, 1.8 to 3.5) (Appendix A), although the age-standardized incidence rate per 100,000 people revealed a different pattern because Sinaloa (85.2, 56.7 to 118.1) and Baja California Sur (74.9, 49.7 to 104.0) had the highest estimates (Figure 1a). Mexico (908, 680 to 1368) and Mexico City (809, 586 to 1120) had the highest estimates for death counts due to PC (Appendix A), and the age-standardized mortality rate ranged from 29.8 (23.7 to 46.9) to 14.2 (10.2 to 24.2) deaths per 100,000 people in Guerrero and Tlaxcala, respectively (Figure 1b). Among the states, Oaxaca and Chiapas showed the highest CFR in 2019 (46.6 and 45.9%, respectively), whereas Baja California Sur and Sinaloa had the lowest CFR in 2019 (25.6 and 25.9%, respectively; Appendix A).

### 3.3. National Burden of Prostate Cancer by Age Group

In Mexico, with an estimated 5194 (3.9 to 7.2 thousand) new incident cases in 2019, the age group of 70 to 74 years represented the peak of PC incidence, whereas this neoplasm caused the highest mortality in the age group of 75 to 79 (1.8, 1.4 to 2.5 thousand) (Figure 2a, Appendix A). However, the count rates per 100,000 showed a contrasting pattern of the burden of PC because the higher estimates of incident cases (564.2, 431.9 to 821.5) peaked in the age group of 85 to 89 years. In contrast, with an estimated 817.9 (569.6 to 1172) deaths per 100,000 people, the group aged >95 years was the most affected by PC (Figure 2a). The age groups of 60–69 and 70–79 years accounted for 69.0% of the new incident cases that were estimated in 2019 for males from 40 to >90 years. In contrast, the highest percentage of deaths due to PC in Mexico occurred among the age groups of 70–79 and 80–89 years (68.6%) (Figure 2b). Finally, the numbers of annual deaths for five-year age groups both consistently increased and remained steady proportionally (Figure 2c).

### 3.4. National Trends of Prostate Cancer Burden from 2000 to 2019

The burden of PC showed a consistent increasing trend in terms of the numbers of incident cases and deaths over the studied period for males of all ages (Figure 3a). However, the magnitude of change differed between these metrics during the last decade, from 2010 to 2019 (Table 1). The incidence of PC for all ages increased from 11.4 thousand (8.9 to 14.0 thousand) in 2000 to 17.8 thousand (14.3 to 23.4 thousand) cases in 2010; and there was a further increase of 52.4% (24.0 to 88.0) in the incident of cases from 2010 to 2019. From 2000 to 2010, the number of deaths increased from 4.8 thousand (3.8 to 6.1 thousand) to 6.9 thousand (5.7 to 9.1 thousand), and then increased 34.8% (13.0 to 62.0) from 2010 to 2019.

At the national level, the trends of the age-standardized rates showed a contrasting pattern (Figure 3b). The incident cases of PC per 100,000 people increased from 43.8 (34.2 to 53.7) at the beginning of the period to 47.1 (38.1 to 60.0) by 2010; there was a further increase by 11.2% (−9.5 to 36.4) from 2010 to 2019. In contrast, the age-standardized deaths per 100,000 people showed a slight reduction, as the death rate values passed from 20.5 (15.9 to 26.0) in 2000 to 19.5 (16.2 to 25.8) in 2010. However, no further reduction was observed from 2010 to 2019 (−0.88, −16.6 to 18.7).

### 3.5. Subnational Trends of Prostate Cancer Burden from 2000 to 2019

At the subnational level, the temporal change in the burden of PC showed both a heterogeneous spatial pattern, as evidenced in the maps depicted in Figure 1**,** and a contrasting trend according to the time series presented in Figure 3b. Regarding the age-standardized incidence per 100,000 people, except for Guerrero, Tabasco, and Campeche, the remaining states followed the national trend and increased during the period from 2010 to 2019; It ranged from 26.8% (−5.5 to 73.2) in Baja California Sur to 4.4% (−20.3 to 37.6) in Oaxaca (Figure 3c). In contrast, 11 states followed the national reduction in age-standardized deaths. The highest reductions were estimated for Guerrero (−15.8, −28.8 to 3.4), Sonora (−5.8, −26.2 to 16.6), and Tabasco (−5.7, −26.1 to 17.8) (Figure 3c).

### 3.6. National Annual Percentage Changes of Age-Standardized Incidence and Mortality of Prostate Cancer from 2000 to 2019

As summarized in Appendix A, according to jointpoint regression analysis, the age-standardized incidence of PC showed a significant AAPC increase by 0.9% (95% CI: 0.7 to 1.2) from 2000 to 2019. The time interval was segmented into three periods comprising 2000–2003, 2003–2016, and 2016–2019, all of which showed a significant APC that ranged between 0.4 and 2.7%; the last segment had the largest change (Appendix A). In contrast, the ASR of deaths due to PC showed a negative and significant AAPC decrease in −0.3% (−0.5 to −0.1) for the period, which was segmented into two periods (2000–2007 and 2007–2019), both of which showed significant decreases (Appendix A).

### 3.7. Association of the Sociodemographic Index with the Burden of Prostate Cancer at the Subnational Level in 2019

According to Appendix A, only the age-standardized incidence per 100,000 people showed a moderate association with SDI (r = 0.42, *p* = 0.008) among the localities in Mexico during 2019, suggesting that in localities with higher values of SDI, there was a higher incidence of PC. In contrast, the rate of PC death at each locality in Mexico was not associated with the SDI (r = 0.11, *p* > 0.05).

## 4. Discussion

Our results revealed that PC caused a higher burden of disease in Mexican males than the global reference. Although PC is linked to high mortality in this population, the national mortality counts and death rate for this neoplasm are lower than those of Argentina, Chile, Cuba, Brazil, Uruguay, and Venezuela, which are some of the countries with the highest mortality rates in the Americas [15]. Moreover, the high mortality rate of PC estimated for Mexico might be partially associated with a delay, since patients affected by this malignancy are frequently diagnosed at advanced stages of the disease [16]. Interestingly, improvements in the management of other cancer types have reduced their mortality trends [15]; however, this behavior has not been observed for PC in Mexico, as its mortality has increased over time. The higher incidence registered in Mexico for PC during recent years might be associated with both the increased use of PSA testing and the increased exposure of individuals to potential risk factors, such as dietary fat and obesity [17,18]. Although the increase in PC incidence has also been related to deficient diagnostic techniques [19,20], the lack of a malignant tumor follow-up registry and population-based cancer registries in Mexico prevents the opportunity to collect relevant clinical information that permits a better understanding of PC epidemiology [5,10].

The national pattern of PC by age showed that the death counts peaked in the 75–79 age group, whereas the incident cases were the highest in the 70–74 age group during 2019. Overall, our series of results concurs with the global pattern, which shows a strong association between older age and an increased risk for developing PC [21,22]. Additionally, we found that young males under 65 years of age were also affected by PC, although in a reduced magnitude when compared to groups >70 years. This result agrees with the global trend of the burden of PC [23], according to which the proportion of patients <65 years affected by this malign neoplasm has increased over the years, possibly indicating novel alterations in the etiology and pathogenesis of the disease.

With respect to the age-standardized rates, we found a contrasting pattern, since the incidence increased while the rate of death decreased during the same period. Likewise, the subnational estimates followed a similar trend for the age-standardized rates, which were also characterized by a heterogeneous geographical pattern of change. These series of results are consistent with a previous study in which the authors assessed the burden of PC at the global level in 195 countries and regions and found not only similar trends in the age-standardized rates of incidence and mortality across all geographic regions but also broad geographical variability [23]. Previous studies have suggested that differences in the incidence and mortality of PC may be associated with disparities in the exposure to risk factors, the detection rate, and the quality of and access to oncology care [19,24,25]. Therefore, some of these factors might partially help explain the subnational disparity observed in our study. Moreover, the incidence and mortality of PC among males of different ages might be influenced by location-specific differences in the medical condition and the exercise and dietary habits of the population; thus, it is necessary to further study their contributions in population-based research.

A recent study by Baade et al. [26] reviewed the geographic disparities linked to differential outcomes for PC in several countries across the globe. The authors found a contrasting pattern of the disease between urban and rural areas because in the former, PSA testing was common, survival was higher, and access to health care facilities was greater in comparison to the latter, where mortality due to PC was higher mostly because diagnoses of the disease were given at advanced stages. Therefore, sociodemographic determinants are relevant for the burden of PC, and Mexico is characterized by disparate socioeconomic determinants that have been linked to the burdens of other types of cancer [9] and some infectious diseases [27]. Although a previous study found an association between the SDI and the burden of PC in 195 countries [23], we only found a moderate association of the SDI with the age-standardized rate of incidence. Nonetheless, our results showed that Chiapas and Guerrero, which are states characterized by the lowest values of SDI in Mexico, had higher rates of mortality, which may be associated with their low educational level, lack of access to health care facilities, and higher proportion of indigenous populations for whom communication is difficult. In conjunction, all these factors make the diagnosis and treatment of PC difficult. Further studies that assess the specific contributions of the socioeconomic determinants among the states of Mexico are needed for understanding this association.

Taken together, our results demonstrated both geographic and temporal variation in the burden of PC among the states of Mexico. Previous reports have suggested that such heterogeneity reflects differences in disease awareness, diagnostic practices, certification of cause of death, and access to health care [25]. Despite the need for future research assessing the potential factors that drive the distinct subnational burdens of PC documented herein, our results emphasize the need for developing and enhancing an adequate health system in Mexico that properly addresses the health problems of male patients >55 years of age, who are prone to develop not only PC but also other malignant tumors. The finding that PC was the leading cause of death in males due to neoplasms should encourage the National Health Ministry to develop more preventive campaigns and effective and extensive diagnostic efforts for the early detection of PC.

## 5. Limitations

Although we used reliable and accurate data from the most recent iteration of the GBD, the input data sources used to calculate the estimates varied both in completeness and quality among the states of Mexico; consequently, this heterogeneity may increase the risk of bias of the estimates. In addition, the lack of data input sources for some years and locations used during the modeling process may contribute to a potential lack of representativeness for some estates, especially in those localities with lower SDI values, where data might be less available or reliable. Therefore, in cases where high-quality data are limited, the estimates rely on covariates and modeling parameters, causing over- or underestimation of the true cancer burden [3]. Thus, in some cases, the estimates should be taken with caution; however, it is better to estimate with a high level of uncertainty than to not and assume that there is no burden of disease at all. This limitation of data reinforces the need for developing and enhancing cancer surveillance and the registry [28]. Finally, the GBD only includes tobacco consumption as a potential risk factor for PC; thus, the absence of other potential risk factors, such as local availability of PSA screening, management of diagnosis and treatment, and other clinical features of the disease, preclude us from carrying out further analysis of the risk factors for PC.

## 6. Conclusions

To the best of our knowledge, this study represents the most updated and comprehensive report for Mexico that describes the burden and trends of PC at the national and subnational levels from 2000 to 2019. Our results demonstrated that PC has caused a significant burden of disease, as this malign neoplasm was among the main causes of death in men during 2019. In addition, the trends described in our study for the incidence and mortality due to PC are similar to those reported for other countries. In addition, we also found subnational variation in the burden of PC. Our results confirm the need for updated and timely estimates of PC because understanding the epidemiological trends and disparities in PC at the subnational level may guide efficient health care planning and adequate resource allocation aimed at increasing disease screening and treatment.

## Figures and Tables

**Figure 1 cancers-14-03184-f001:**
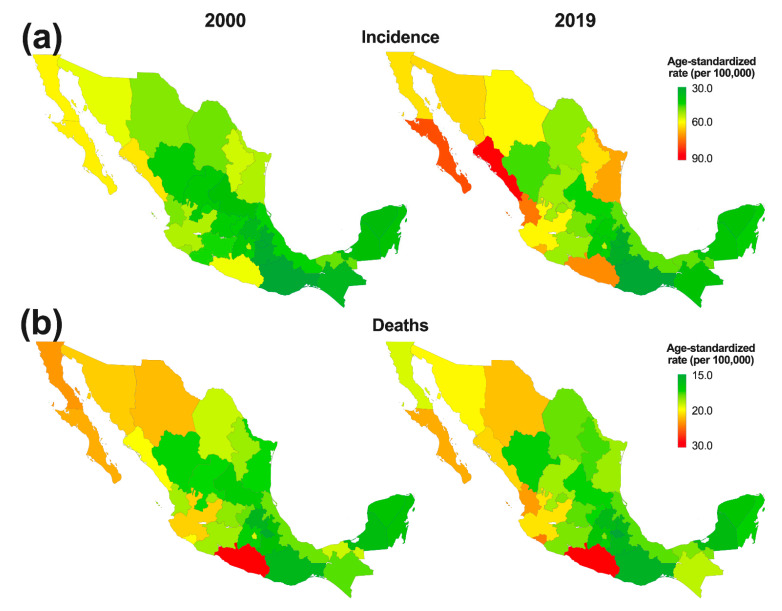
Change in the subnational age-standardized (**a**) incidence and (**b**) deaths per 100,000 people due to prostate cancer in Mexico from 2000 to 2019.

**Figure 2 cancers-14-03184-f002:**
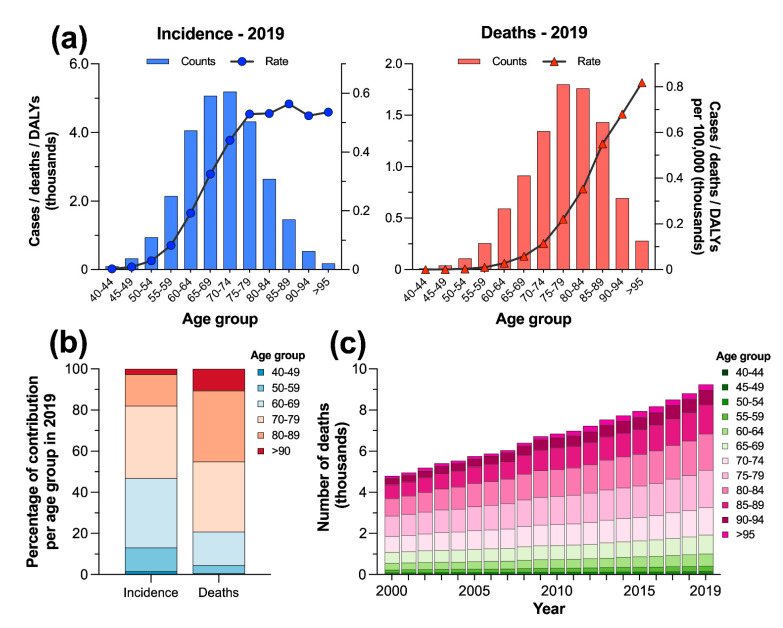
(**a**) Prostate cancer burden by age group and rate per 100,000 people in Mexico in 2019; (**b**) relative contribution per age group to the counts of incident cases and deaths; and (**c**) contribution per age group to the annual number of deaths due to prostate cancer in Mexico from 2000 to 2019.

**Figure 3 cancers-14-03184-f003:**
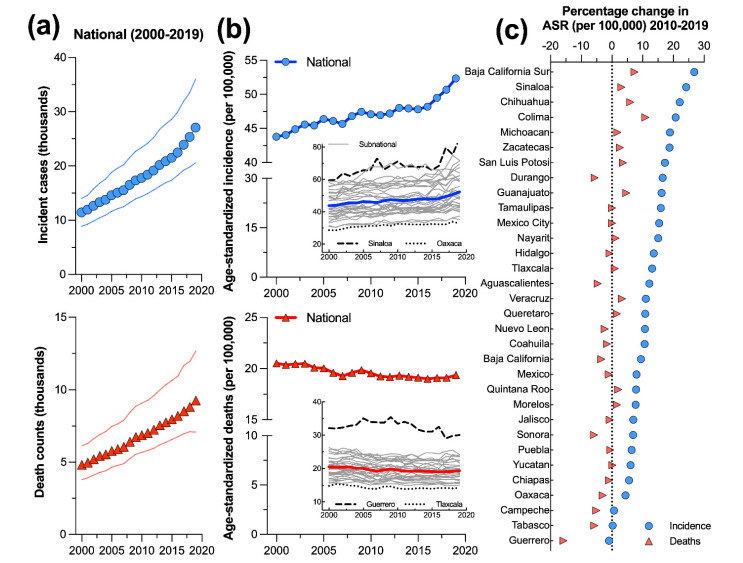
National trends of (**a**) prostate cancer incidence and death counts; (**b**) age-standardized rates per 100,000 people in Mexico from 2000 to 2019; and (**c**) percentage of change in the burden of prostate cancer from 2010 to 2019 at the subnational level. In (**b**), the insert shows the subnational trends, including the names of the states with the highest and lowest rates for 2019.

**Table 1 cancers-14-03184-t001:** Estimates of prostate cancer burden in Mexico.

Estimate	Counts (95% UI)			% Change 2010 to 2019	ASR (per 100,000)
	2000	2010	2019		2019
Incidence	11,444 (8883 to 14,016)	17,784 (14,346 to 23,397)	27,096 (20,602 to 36,016)	52.4 (24.0 to 88.0)	52.3 (40.0 to 70.1)
Deaths	4812 (3795 to 6138)	6864 (5674 to 9075)	9256 (7077 to 12,678)	34.8 (13.0 to 62.0)	19.4 (14.7 to 26.7)

ASR = age-standardized rate.

## Data Availability

The datasets analyzed in the current study are available at the GHDx website (http://ghdx.healthdata.org/gbd-2019/data-input-sources accessed 15 January 2022). In addition, all the estimations used to perform the analyses and the figures are available from the corresponding author.

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
