# Peer review of "Incidence, Mortality, and Trends of Prostate Cancer in Mexico from 2000 to 2019: Results from the Global Burden of Disease Study 2019"

_cancers, 2022, doi:10.3390/cancers14133184_

Round 1

Reviewer 1 Report

This is well designed analysis that describes the trends and burden of PC between 2009-2019, in Mexico at a national and sub-national level, reporting PC deaths among the main causes of death in Mexican men, in 2019. The authors have acknowledged the incompleteness f data may increase risk of bias which is a critical in the interpretation of results. As such it would e useful if they could elaborate on this risk and how may it have biased outcome.

Author Response

We thank the associate editor and reviewer’s thoughtful comments and valuable suggestions which help us improve the quality of the paper. We have revised our manuscript according to the associate editor and reviewer’s comments and here we present the response to each one of those comments.

  1. This is a well designed analysis that describes the trends and burden of PC between 2009-2019, in Mexico at a national and sub-national level, reporting PC deaths among the main causes of death in Mexican men, in 2019. We thank the reviewer by the thoughtful feedback and the support to our manuscript, which intends to provide useful information to measure the magnitude of the health problem caused by PC in Mexico.

  1. The authors have acknowledged the incompleteness of data may increase risk of bias which is a critical in the interpretation of results. As such it would be useful if they could elaborate on this risk and how may it have biased outcome. We thank the reviewer for this important suggestion. We have now included the following text:he manuscript, starting from the abstract to the discussion.

” In addition, the lack of data input sources for some years and locations used during the modeling process may contribute to a potential lack of representativeness of the GBD estimates for some estates, especially in those localities with the lower SDI values, where data might be less available or reliable. Therefore, as previously stated in a study assessing the global burden of cancer [3], in cases where high-quality data is limited for some years and locations, the estimates rely on covariates and modeling parameters, thus causing an over- or underestimation of the true cancer burden. Therefore, in some cases the estimates should be taken with caution, though it is better to estimate with a high level of uncertainty, that not estimating and assume that there is no burden of disease at all. This limitation of data reinforces the need for developing and enhancing cancer surveillance and registry [12].”

Reviewer 2 Report

Thank you very much for your work. It is a comprehensive study to reflect the epidemiology of prostate cancer in Mexico. Authors may consider the following before it is considered to be published in Cancers. Points in bold are my major concerns.

  1. Line 157. For trend analysis, apart from simple computation of percentage change from just two timepoints (2000 and 2019), authors should also report annual percentage change (APC) across 2000-2019 with joinpoint regression, followed by annual average percentage change (AAPC) (plus 95% CI, P value) which is estimated to figure out the trend direction and degree by considering the slope and weighting of each segment. Authors may refer to a recently published paper in Cancers for ovarian cancer: https://doi.org/10.3390/cancers14092230. This  applies to both national and subnational data.
  2. Line 168-169. Any reference for the use of Spearman correlation?
  3. Line 169. Authors should specify what software was used for data analysis, apart from Prism for figures.
  4. Suggest adding subtitles in Methods and Results to clearly indicate the part of incidence, mortality, DALYs, etc instead of just “PC burden”.
  5. Methods – Please briefly describe the analysis stratified by age in the Methods.
  6. Authors discussed that Mexico is a distinct geographical area. Authors should also provide more information about the population size and its changes across the study period in both Introduction and Discussion, so as to reflect that its residential mobility is stable (by some simple statistical analysis if any), which should not bring great influence to incidence and mortality.
  7. Line 303-305. I think these sentences are from the template?
  8. Line 383. “a weak association of the SDI with the age-standardized rate of incidence.” Suggest modifying the phrase of “weak” to “moderate”.
  9. Authors are also suggested to discuss more about the implications of the study results e.g. potential articulations, policy-making, decision making, healthcare and social system.

Author Response

We thank the associate editor and reviewer’s thoughtful comments and valuable suggestions which help us improve the quality of the paper. We have revised our manuscript according to the associate editor and reviewer’s comments and here we present the response to each one of those comments.

"Please see attached file"

  1. Thank you very much for your work. It is a comprehensive study to reflect the epidemiology of prostate cancer in Mexico. We thank the reviewer for the support to our study and the thoughtful comments and suggestions that improved our manuscript.

  1. Line 157. For trend analysis, apart from simple computation of percentage change from just two timepoints (2000 and 2019), authors should also report annual percentage change (APC) across 2000-2019 with joinpoint regression, followed by annual average percentage change (AAPC) (plus 95% CI, P value) which is estimated to figure out the trend direction and degree by considering the slope and weighting of each segment. Authors may refer to a recently published paper in Cancers for ovarian cancer: https://doi.org/10.3390/cancers14092230. This  applies to both national and subnational data. We thank the reviewer for this important suggestion. The manuscript now includes the following:

2.6. Analysis of prostate cancer trends

Additionally, we perfomed a jointpoint regression analysis to evaluate the temporal trend in the national ASR of incidence, mortality, and DALYs using the Jointpoint regression software 4.9 available online (https://surveillance.cancer.gov/joinpoint/) from the National Cancer Institute. The model finds segments (time intervals) based on best-fitting points identified by the slope of the changing trend for the segment, then connects a set of statically linear models on a logarithic scale [13]. Based on ASR and their standard errors, the model calculates the annual percent change (APC) for any given segment and expresses the estimate assuming a constat rate during the time interval. Additionally, the analysis calculates the average annual percent change (AAPC) for the entire time interval. APC and AAPC were estimated with 95% Confidence Interval (95% CI) to assess both the direction and magnitude of change considering the value and sign of the slope for a given segment or the full period, respectively [14]. Finally, we assessed whether APC and AAPC were significantly different from zero by using a two/sided t-distribution test at the alpha level=0.05.

Results

As summarized in Table 2, according to jointpoint regression analysis, the age-standardized incidence of PC showed a significant average annual increase of 0.9% (95% CI: 0.7 to 1.2) from 2000 to 2019. Besides, the time interval was segmented into three periods comprising 2000-2003, 2003-2016, and 2016-2019; all of which showed a significant APC that ranged between 0.4 to 2.7%, being the last segment the one with the highest estimated annual change (Supplementary Figure 2a). On the contrary, both the age-standardized rate of deaths and DALYs due to PC showed a negative and significant average annual decrease of -0.4% (95% CI: -0.6 to -0.3) and -0.3% (-0.5 to -0.1) for the period, respectively (Table 2). However, in these two metrics there were only two segments that varied both in the time interval and the direction and magnitude of the APC. While the age-standardized rate of deaths showed a significant annual decrease for the two segments (Supplementary Figure 2b), the APC of the DALYs showed an increase between 2016 and 2019 (Supplementary Figure 2c).

Table 2. Estimates of Average Annual Percent Change and Annual Percent Change of the age-standardized rate of incidence, deaths, and DALYs (per 100,000 people) due to prostate cancer burden in Mexico.

Estimate

Metric

Incidence

Range

Period

AAPC (95% CI)

Test Statistic (t)

Prob > |t|

Full range

2000 - 2019

0.9 (0.7 to 1.2)*

7.0

< 0.001

Segment

Period

APC (95% CI)

Test Statistic (t)

Prob > |t|

1

2000 - 2003

1.4 (0.2 to 2.6)*

2.6

0.022

2

2003 - 2016

0.4 (0.3 to 0.6)*

6.6

< 0.001

3

2016 - 2019

2.7 (1.4 to 4.0)*

4.5

0.001

Deaths

Range

Period

AAPC (95% CI)

Test Statistic (t)

Prob > |t|

Full range

2000 - 2019

-0.4 (-0.6 to -0.3)*

-5.2

< 0.001

Segment

Period

APC (95% CI)

Test Statistic (t)

Prob > |t|

1

2000 - 2007

-0.8 (-1.1 to -0.4)*

-4.5

< 0.001

2

2007 - 2019

-0.2 (-0.4 to -0.0)*

-2.7

0.015

DALYs

Range

Period

AAPC (95% CI)

Test Statistic (t)

Prob > |t|

Full range

2000 - 2019

-0.3 (-0.5 to -0.1)*

-2.6

0.010

Segment

Period

APC (95% CI)

Test Statistic (t)

Prob > |t|

1

2000 - 2016

-0.5 (-0.6 to -0.4)*

-10.7

< 0.001

2

2016 - 2019

0.9 (-0.4 to 2.2)

1.4

0.178

AAPC: Average Annual Percent Change; APC: Annual Percent Change

* Indicates that the APC/AAPC is significantly different from zero at p < 0.05.

Supplementary Material

Supplementary Figure 2. Results from jointpoint regression analysis of (a) incidence, (b) deaths, and (c) DALYs of prostate cancer in Mexico from 2000 to 2019.

  1. Any reference for the use of Spearman correlation? Thanks to the reviewer for this important suggestion. We corrected this and now included the following information to the manuscript:

”Based on results from Shapiro-Wilk test for normal distribution (Incidence, W=0.96, p=0.393; DALYs, W=0.943, p=0.095; and Deaths, W=0.954, p=0.189), we performed a simple Pearson correlation analysis to assess the linear association between the subnational SDI and each metric.”

  1. Line 169. Authors should specify what software was used for data analysis, apart from Prism for figures. Thank you for this comment. We included more details regarding the specific software used for the different analysis. The text now includes the following:

“Data processing and analyses were conducted using Python, version 3.7.0 (Python Software Foundation); Stata, version 15.1 (StataCorp); and R, version 3.4.1 (R Foundation). Code is available at https://ghdx.healthdata.org/gbd-2019/code [3].

Additionally, we perfomed a jointpoint regression analysis to evaluate the temporal trend in the national ASR of incidence, mortality, and DALYs using the Jointpoint regression software 4.9 available online (https://surveillance.cancer.gov/joinpoint/) from the National Cancer Institute.”

  1. Suggest adding subtitles in Methods and Results to clearly indicate the part of incidence, mortality, DALYs, etc instead of just “PC burden”.Thank you for this recommendation. We added the following subheadings and separated the corresponding paragraphs to enhance the readability and understanding of each modeling step to produce the estimates. The text now includes the following:

2.3. Estimation of prostate cancer incidence and mortality

A detailed description of the PC incidence and mortality estimates for the GBD have been previously published in a paper focused on 29 cancer groups [15]. The modeling framework and a detailed flowchart including specific codes for the cancer burden estimation in the GBD are provided in the following web address: http://ghdx.healthdata.org/gbd-2019/code/cod-2. In brief, as described previously in a study of gastric cancer that used GBD estimates [16], the modeling steps included 1) calculating the mortality-to-incidence ratio (MIR) using data sources that included incidence and mortality for PC, 2) collecting the incidence of PC for each cancer registry, 3) estimating PC mortality by multiplying the incidence data by the corresponding MIR, 4) including PC mortality sets into the PC cause of death database using the Cause of Death Ensemble model (CODEm) and processing the data to estimate the cancer-specific mortality of PC, and 5) calculating the incidence of PC by the estimated cancer-specific mortality of PC and MIR.

2.4. Estimation of DALYs due to prostate cancer

The disability-adjusted life-years (DALYs) incorporate both the fatal and nonfatal burden of PC and correspond to the sum of years lived with disability (YLDs) and years of life lost (YLLs) [17]. To estimate YLDs, the disability weight (ranging from 0 “no health loss” to 1 “dead”) caused by a specific sequelae disease is multiplied by the prevalence of such disease, whereas the YLLs represent the sum of years of life lost due to premature mortality multiplied by the standard life expectancy [18].

3.1. National incidence, mortality, and DALYs of prostate cancer in 2019

In Mexico, there were 27.1 thousand (95% UI, 20.6 to 36.0 thousand) incident cases of PC and 9.2 thousand (95% UI, 7.7 to 12.7 thousand) deaths in 2019 for men of all ages. By the same year, the age-standardized incidence and mortality rates were estimated at 52.3 (40.0 to 70.1) cases and 19.4 (14.7 to 26.7) deaths per 100,000 people, respectively. In addition, PC was estimated to cause 161.5 thousand (122.7 to 219.5 thousand) DALYs in 2019, with an age-standardized rate estimated at 319.5 (244.1 to 432.5) DALYs per 100,000 people (Table 1).

3.2. Subnational incidence, mortality, and DALYs of prostate cancer in 2019

At the subnational level, the highest number of incident cases in 2019 occurred in the state of Mexico (2.8 thousand, 2.1 to 4.2) and Mexico City (2.5 thousand, 1.8 to 3.5) (Supplementary Table 1). The age-standardized incidence rate showed a different pattern because with 85.2 (56.7 to 118.1) and 74.9 (49.7 to 104.0) new cases per 100,000 people, Sinaloa and Baja California Sur had the highest estimations (Figure 1a). The number of deaths of men of all ages due to PC showed the highest estimates in Mexico (908, 680 to 1,368) and Mexico City (809, 586 to 1,120) (Supplementary Table 2). The age-standardized mortality rate ranged from 29.8 (23.7 to 46.9) to 14.2 (10.2 to 24.2) deaths per 100,000 people in Guerrero and Tlaxcala, respectively (Figure 1b). Finally, the estimation of DALYs ranged from 14.4 thousand (10.3 to 19.7 thousand) in Mexico to 1.0 thousand (0.67 to 1.4 thousand) in Campeche (Supplementary Table 3), and the age-standardized rate was the highest in Guerrero, where PC caused 487.4 (391.4 to 718.2) DALYs per 100,000 people (Figure 1c).

3.5. National annual percent change of age-standardized incidence, mortality, and DALYs of prostate cancer from 2000 to 2019”

  1. Methods – Please briefly describe the analysis stratified by age in the Methods. We thank the reviewer for this important suggestion. The following text has been added to the manuscript:

2.7. Analysis of prostate cancer burden by age group

To assess the specific pattern of PC burden according to the age groups, only estimates from 40 to >95 years were included. Crude and ASR (per 100,000 people) estimates for the incidence, mortality, and DALYs due to PC per age group of five years (40-44, 45-49, 50-54, 55-59, 60-64, 65-69, 70-74, 75-79, 80-84, 85-89, 90-94, and >95 years of age) are summarized in Supplementary Table 4. Besides, using these groups, we plotted the death counts by year from 2000 to 2019 to assess the specific contribution of each group to the mortality due to PC in Mexico. Finally, we compacted age groups into the following categories: 40-49, 50-59, 60-69, 70-79, 80-89, and >90 years of age to assess their relative contribution to the incidence, mortality, and DALYs counts in 2019.”

  1. Authors discussed that Mexico is a distinct geographical area. Authors should also provide more information about the population size and its changes across the study period in both Introduction and Discussion, so as to reflect that its residential mobility is stable (by some simple statistical analysis if any), which should not bring great influence to incidence and mortality. We thank the reviewer for attracting attention into this important issue. In the present study, we intentionally avoided including information regarding the changes in the national population and the trends among the 32 states of Mexico. The main reason for this is that we are working on a study that assesses the burden of 29 groups of cancer in which we include a detailed comparison regarding the population changes split by sex and type of cancer. Therefore, we kindly ask the reviewer to wait a little bit for us to complete this study. Meanwhile, we provide the reviewer with the following two paragraphs that will be incorporated in this future study:

 “According to the 2020 Census, a total population of 126,014,024 people reside in Mexico. Mexico ranks 11th as the most inhabited Country in the world. During the last 70 years its population has quadruplicated, growing from 25.8 million in 1950, to 126 million in the year 2020. In the last decade, the population has increased by 14 million of inhabitants, with places such as Mexico and Jalisco at the top of the most populated Mexican States [19].

As well important as population growth is the change in average age of the total population. From year 2000 to 2020, the average age increased from 26 to 29 years [19]. Thus, if this trend carries on, a serious problem for the health system is to come. Furthermore, the elder population is expected to double for year 2050, increasing from 8.8% to 18.6% [20]. Also on the rise, is the number of deaths caused by cancer, with malignant tumors recognized as the main cause. In 2020, compared to year 2000, the number of registered deaths had a significant increase, reaching over a million. Compared to 2019, where the total number of deaths was 747 thousand, an increase of just over 300,000 deaths was registered [19].”

  1. Line 303-305. I think these sentences are from the template We thank the reviewer for pointing out this mistake. The lines were removed from the manuscript.

  1. Line 383. “a weak association of the SDI with the age-standardized rate of incidence.” Suggest modifying the phrase of “weak” to “moderate”.The text has been corrected accordingly.

  1. Authors are also suggested to discuss more about the implications of the study results e.g. potential articulations, policy-making, decision making, healthcare and social system. The following text was added to the manuscript:

” Taken together, our results demonstrated both a geographic and temporal variation in the burden of PC among the states of Mexico. Previous reports have suggested that such a heterogeneity might reflect differences in disease awareness, diagnostic practices, certification of cause of death, and access to health care [10]. Despite the need for future research assessing the potential factors that drive the differential subnational burden of PC documented herein, our results emphasize the need for developing and enhancing and adequate health system in Mexico that properly addresses the health problems of male patients >55 years of age, which are prone to develop not only PC but other malignant tumors. The finding that PC was the leading cause of death in males due to neoplasm should encourage the National Health Ministry to develop more aggressive preventive campaigns as well as effective and extensive diagnostic efforts for early detection of PC cases thorough the territory.”

Round 2

Reviewer 2 Report

Authors have addressed all of my concerns in a satisfactory way.